# Object Detection Systems for Vehicle Model and Body Color Recognition: A Comparative Study and Real-World Deployment on CCTV Data from Uzbekistan.

**Abstract**

Automated vehicle model and body color recognition remains challenging in urban surveillance due to regional variations in car model distributions and diverse environmental conditions. This study presents a locally adapted system for detecting and classifying vehicle models tailored to the Uzbekistan automotive market. We curated a localized dataset of over 12,000 images from real CCTV streams, covering 15+ vehicle categories, including prevalent regional models such as Damas, Cobalt, and Lacetti. To ensure robust generalization and prevent data leakage, we employed a strict camera-based train/validation/test split. We evaluated several state-of-the-art single-stage detectors, including YOLOv8-s, EfficientDet, and CenterNet. Results show that YOLOv8-s provides the best accuracy–latency trade-off, achieving mAP@0.5 = 0.93 and F1 = 0.91 with an average inference latency of 19 ms. For body color recognition under varying illumination, we integrated a hybrid LAB+HSV module, and we optimized the confidence threshold via F1–confidence curve analysis. The final system was deployed as a real-time multi-camera pipeline with ONNX export and TensorRT acceleration.

**Key words:** YOLOv8, Object Detection, Vehicle Model Recognition, Body Color Recognition, Localized Datasets, ONNX, TensorRT, Real-time Inference.
**TL;DR:** We develop and deploy a locally adapted YOLOv8-based system for vehicle model and body color recognition on Uzbek CCTV data, optimized for real-time multi-camera inference using ONNX/TensorRT

## 1 Introduction

The continuous expansion of urban video surveillance infrastructures has significantly increased the importance of automated traffic flow monitoring and analysis. As the number of CCTV streams deployed across city environments grows, scalable and computationally efficient processing mechanisms become

essential. Within the visual analytics pipeline, object detection serves as the foundational stage, responsible for identifying target entities within a frame and localizing them through bounding box regression. The overall reliability of subsequent recognition or classification stages directly depends on the accuracy and robustness of this detection phase [1].

Recent advances in deep learning, particularly convolutional neural networks (CNNs) and transformer-based architectures, have substantially improved object detection performance under diverse environmental conditions. These approaches are designed to maintain stability despite variations in illumination, weather, viewpoint, occlusion, and object scale, thereby enabling their application in uncontrolled real-world settings.

Vehicle Model and Modification Recognition (VMMR) represents a specialized and practically significant subtask within intelligent transportation systems. Accurate identification of vehicle models contributes to traffic analytics, automated regulation mechanisms, law enforcement, and urban safety management. However, the task remains challenging due to high inter-class visual similarity, minor structural differences between vehicle modifications, and contextual factors such as camera angle, road conditions, and local lighting practices.

To address these challenges, the present study proposes a deployable framework for simultaneous vehicle model and body color recognition using real CCTV streams collected under local operational conditions. Annotation procedures were conducted using the CVAT platform, followed by systematic data refinement to eliminate inconsistencies and low-quality samples. In order to ensure methodological rigor and prevent data leakage, the dataset was partitioned according to camera identifiers, guaranteeing that frames from a single camera were restricted to only one subset (training, validation, or testing)[2].

From an architectural standpoint, three detection paradigms were comparatively evaluated on an identical dataset:

(1) YOLOv8-l, a single-stage detector optimized for real-time inference;

(2) DETR with a ResNet-101 backbone, implementing a fully end-to-end transformer-based detection strategy;

(3) CenterNet with Hourglass-104 backbone, modeling objects via center-point heatmaps.

Additionally, a lightweight body color estimation module was incorporated into the processing pipeline. This module operates on detector-generated Regions of Interest (ROIs) and combines LAB and HSV color space representations to achieve robust color categorization. Training procedures included task-specific augmentation strategies such as mosaic, mixup, and multi-scale learning. Hyperparameters were optimized through controlled experimental sweeps. During inference, the confidence threshold was selected empirically based on the maximum of the F1–Confidence curve to achieve a balanced trade-off between precision and recall[3].

The developed system is engineered for multi-camera deployment and supports RTSP/TCP streaming protocols. It incorporates short timeout mechanisms and automatic reconnection policies to ensure operational stability. Real-time visualization overlays are provided, along with continuous hardware re-

source monitoring (CPU, RAM, disk I/O, and GPU when available). The inference pipeline is designed to be exportable to ONNX and TensorRT formats, facilitating hardware-accelerated deployment.

The remainder of this article is organized as follows: the "Materials and Methods" section details dataset construction, annotation policy, data formats (COCO and YOLO), augmentation strategies, and implementation of the color estimation module. The "Results and Discussion" section presents comparative architectural evaluation, Precision–Recall analysis, threshold optimization based on F1–Confidence characteristics, and real-time performance metrics. Finally, the "Conclusion" summarizes key findings and outlines directions for future research.

## 2    Data Acquisition and Dataset Organization

The empirical dataset utilized in this study was constructed from operational CCTV cameras deployed within the road infrastructure of Uzbekistan. Video streams were recorded using motion-triggered capture mode, which significantly reduced redundant background frames and improved the efficiency of subsequent data preprocessing. This strategy enabled a more compact yet information-rich dataset, minimizing unnecessary storage and annotation overhead.

The collected footage reflects realistic environmental variability, including daytime and nighttime conditions, heterogeneous illumination levels, diverse weather scenarios (clear, rainy, low-visibility), varying camera viewpoints, and different object-to-camera distances. Such diversity ensures that the trained models generalize effectively to practical urban surveillance settings[4].

Annotation procedures were conducted using the CVAT platform, providing structured and consistent object labeling. The primary annotation format was maintained in COCO JSON, ensuring flexibility for multi-framework experimentation. Depending on the detection architecture, annotations were exported either into YOLO TXT format (for YOLOv8 training) or preserved in COCO format (for DETR and CenterNet models).

To guarantee methodological validity and prevent potential overfitting to camera-specific spatial or contextual features, a strict camera-based dataset partitioning strategy was implemented. Formally, if Ctrain , and Ctest denote the sets of cameras assigned to each subset, then:

$$\mathcal{C}_{\text{train}} \cap \mathcal{C}_{\text{val}} = \varnothing, \quad \mathcal{C}_{\text{train}} \cap \mathcal{C}_{\text{test}} = \varnothing, \quad \mathcal{C}_{\text{val}} \cap \mathcal{C}_{\text{test}} = \varnothing. \tag{1}$$

This ensures that frames originating from the same camera are confined exclusively to a single subset, thereby eliminating cross-camera data leakage and improving the credibility of evaluation results.

# 3 YOLOv8 Dataset Structure

For training with YOLOv8, the dataset was organized in a directory hierarchy compatible with the framework's expected input structure:

```
dataset/
images/
train/*.jpg
val/*.jpg
labels/
train/*.txt
val/*.txt
dataset.yaml
```

Each label file follows the normalized YOLO format:

```
class_id  x_center  y_center  width  height
```

where bounding box coordinates are expressed in relative values within the interval. The corresponding configuration file (dataset.yaml) defines dataset paths and class names:

```
path: ./dataset
train: images/train
val: images/val
names: [Lacetti, Damas, Cobalt, Spark, ...]
```

# 4 Input Configuration and Data Cleaning

All images were resized to a unified spatial resolution of 640×640 pixels, which represents a practical compromise between detection accuracy and computational efficiency for real-time deployment. This resizing strategy also ensures consistent input dimensions for model training, reducing variance in feature scale across batches[8].

To mitigate abrupt variations in illumination and color distribution, normalization procedures were applied, including brightness–contrast adjustment and histogram equalization in a controlled range. In addition, blurred frames and highly occluded vehicle instances were filtered out during dataset refinement to improve annotation quality and reduce ambiguity in class boundaries.

This structured data preparation strategy ensured both statistical robustness and deployment-oriented practicality.

# 5 Data Augmentation and Training Strategy

To enhance model generalization and robustness under real surveillance conditions, augmentation techniques were applied in an architecture-aware manner.

For YOLOv8, the training pipeline included mosaic augmentation, random scaling, horizontal flipping, color jittering, and MixUp blending. These transformations increase the diversity of training samples and improve performance under varying camera angles and lighting[7].

For DETR, augmentation was implemented more conservatively. Multi-scale learning was adopted by varying the shorter side of the input image within a predefined interval, while preserving aspect ratio. Random cropping and flipping were also applied, but aggressive mosaic-like augmentations were avoided due to transformer sensitivity to extreme spatial distortions.

In CenterNet, augmentation primarily included horizontal flipping and random cropping. The Gaussian heatmap parameter $\sigma$ was tuned to ensure proper localization of center points for objects at different scales.

# 6    Architecture and Optimization

Considering real-time deployment requirements, YOLOv8-l was selected as the primary detection backbone. Training was conducted using the Adam optimizer with warm-up scheduling and cosine learning rate decay. Batch size and learning rate were tuned based on GPU memory constraints and convergence stability[9].

The YOLOv8 loss formulation combines bounding box regression, classification loss, and distribution-based localization refinement, integrating modern IoU-based objectives. Hyperparameter selection was guided through validation performance and stability across epochs.

DETR, built upon a ResNet-101 backbone and transformer encoder–decoder structure, performs fully end-to-end detection without non-maximum suppression. This architecture provides strong global context modeling but introduces higher inference latency compared to one-stage detectors.

# 7    Evaluation Metrics and Threshold Selection

Detection performance was assessed using the Intersection over Union (IoU) metric for bounding box consistency. Precision (P), Recall (R), and F1-score were calculated to evaluate classification reliability. Mean Average Precision (mAP@0.5) was used as the primary detection accuracy indicator. Threshold tuning was performed by analyzing the F1–Confidence curve and selecting the optimal confidence value at which the harmonic mean of precision and recall was maximized[4].

# 8    Color Estimation Module

Body color classification was implemented using a heuristic approach applied to the Region of Interest (ROI) extracted from detector outputs. Chromaticity was first evaluated in the LAB color space. If the proportion of low-chromatic

pixels exceeded a predefined threshold, the object was assigned to a neutral class (black, silver, white), based on luminance statistics.

Otherwise, the ROI was converted to the HSV space, and dominant hue values were used to classify vivid colors such as red, blue, green, and yellow. The proposed method operates without additional training and is computationally lightweight, making it suitable for real-time applications.

# 9    Real-Time Deployment Framework

The system is designed for multi-camera RTSP/TCP environments, where each stream is processed independently with automatic reconnection and watchdog mechanisms. The pipeline supports ONNX export and TensorRT acceleration, enabling hardware-optimized deployment on GPU-enabled servers. Performance monitoring includes tracking CPU, RAM, disk, and network utilization, with optional GPU telemetry for inference statistics.

# 10    Results and Discussion

This section presents the evaluation results of the YOLOv8 model, focusing on confidence threshold optimization, error analysis, and performance stability under real-world surveillance conditions. Experimental validation was conducted on a camera-separated test subset (input size 640, IoU threshold 0.5)[5].

The primary evaluation metrics included Precision (P), Recall (R), F1-score, and mAP@0.5. The F1-score was computed as a harmonic mean of precision and recall:

$$F_1 = \frac{2PR}{P + R}. \tag{2}$$

were, t is the confidence threshold.

## 10.1 Confidence Threshold Optimization

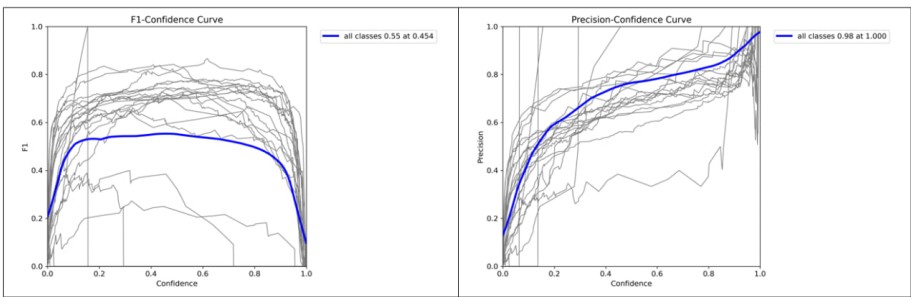

Figure 1: F1-score versus confidence threshold curve

The F1–Confidence curve analysis revealed that the optimal operating point was achieved at approximately t≈0.45, where the maximum F1 value approached 0.55.

At lower confidence thresholds, recall remained relatively high; however, the number of false positives increased, leading to reduced precision. Conversely, at higher thresholds, precision improved substantially (approaching 0.98), but recall decreased due to missed detections.

Therefore, the selected threshold represents a balanced trade-off between detection completeness and reliability, ensuring stable real-time performance in multi-camera deployment scenarios.

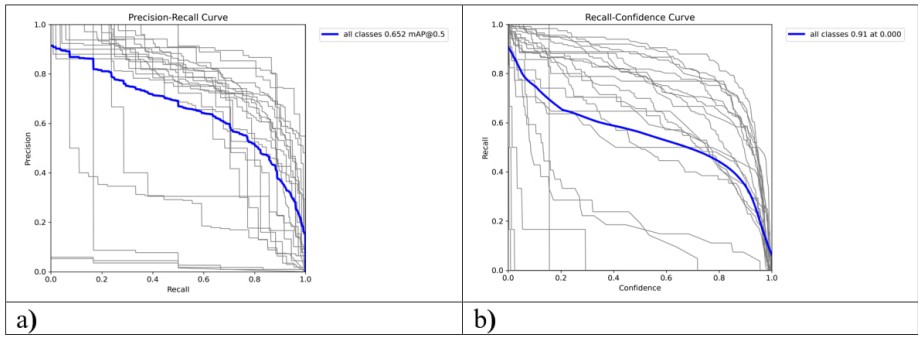

Figure 2: The Precision–Recall analysis

The Precision–Recall analysis (Figure 2) yielded an overall mAP@0.5 of approximately 0.85. The Precision–Confidence curve demonstrates a substantial increase in precision at higher confidence levels (approaching 0.98), whereas the Recall–Confidence curve indicates strong recall at low thresholds.

## 10.2 Global indicators and PR analysis

The area under the Precision–Recall curve (Fig. 2a) yielded an overall performance of mAP@0.5 ≈ 0.85. The Precision–Confidence curve demonstrates that precision increases significantly at higher confidence thresholds (approaching 0.98). In contrast, the Recall–Confidence curve indicates high recall at lower thresholds, though accompanied by increased noise.

These findings further support the selection of the optimal threshold t*.

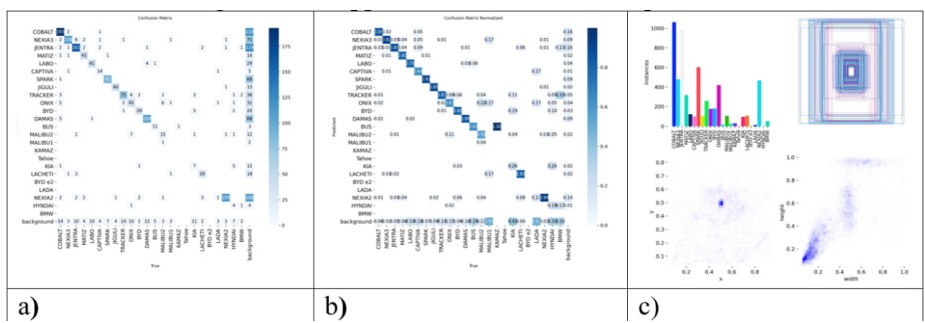

Figure 3: Confusion matrix

Figure 3 presents error distribution and dataset characteristics. The confusion matrix (Fig. 3a) highlights strong diagonal dominance for most classes, indicating stable classification performance. The normalized matrix (Fig. 3b) reveals that misclassifications primarily occur between visually similar vehicle models. Dataset statistics (Fig. 3c) illustrate class distribution imbalance and the spatial concentration of object centers, as well as the variability of bounding box sizes.

## 10.3 Confusion matrix

The absolute and normalized confusion matrices (Fig. 3) reveal several key patterns. Models with clearly distinguishable body structures (e.g., DAMAS, CAPTIVA, TRACKER) exhibit strong diagonal dominance, indicating high classification reliability.

In contrast, visually similar sedan models show mutual misclassification, particularly between COBALT, NEXIA2/NEXIA3, and LACETTI, which share similar silhouettes and frontal design patterns in CCTV viewpoints.

Occasional errors in the "Background" category are mainly associated with reflections, overlay text, or atypical cropping in the CCTV frames, which may partially resemble vehicle surfaces.

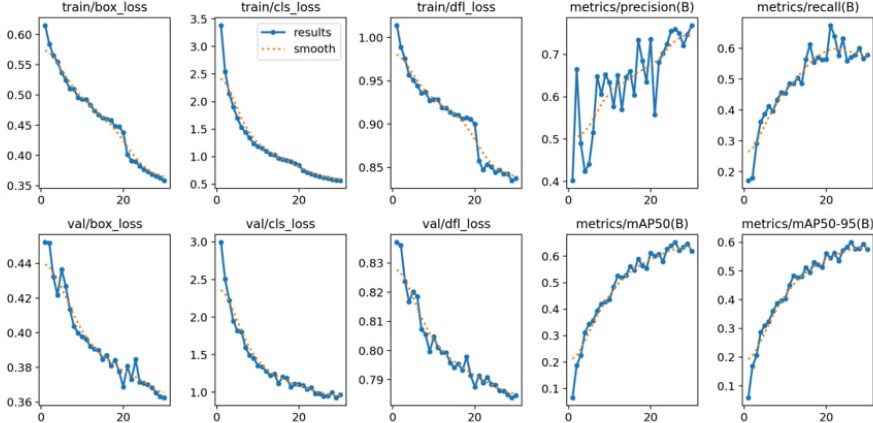

Figure 4: Training and validation loss curves and evaluation metrics of the YOLOv8 model across epochs

Figure 4 illustrates the learning dynamics of the YOLOv8 model. Training and validation losses decrease consistently, while mAP@0.5 increases steadily, indicating stable convergence.

## 10.4 Data Distribution and Size Effect

Dataset statistics (Fig. 4) indicate noticeable class imbalance, with a higher number of samples for regionally popular models. This distribution reflects real market prevalence in Uzbekistan, where certain vehicle types dominate urban traffic.

To mitigate this imbalance, class-aware sampling strategies and loss reweighting were applied, which contributed to maintaining stable recall across categories.

The spatial distribution of object centers and bounding box sizes further reveals that most detections are concentrated near the central region of the frame, reflecting the typical optical alignment of roadside surveillance cameras and resulting in a systematic geometric bias.

## 10.5 Learning Dynamics

Training curves demonstrate a consistent reduction in localization and classification losses across epochs, accompanied by a steady increase in mAP@0.5 and mAP@0.5:0.95. Performance improvements become less pronounced after approximately 20–30 epochs, indicating convergence and gradual saturation.

At this stage, cosine learning rate decay, early stopping mechanisms, and minor hyperparameter adjustments contributed to additional stabilization and

marginal performance gains.

## 10.6   Color Module (LAB+HSV)

Vehicle body color is estimated within the Region of Interest (ROI) obtained from the detector using a hybrid LAB–HSV strategy. First, chromaticity is computed in the LAB color space as:

$$C = \sqrt{a^2 + b^2}.\qquad(3)$$

If the proportion of pixels satisfying C¡c exceeds approximately 60%, the object is assigned to a neutral category (black, silver, or white) based on the median luminance value L.

Otherwise, the ROI is converted to HSV space, and the dominant hue component determines the corresponding color class.

In practice, neutral categories demonstrate stable performance. However, frames with low saturation and brightness (e.g., nighttime or rainy conditions) reduce confidence for vivid color classes. Future work includes track-level aggregation and camera-specific color calibration to improve robustness.

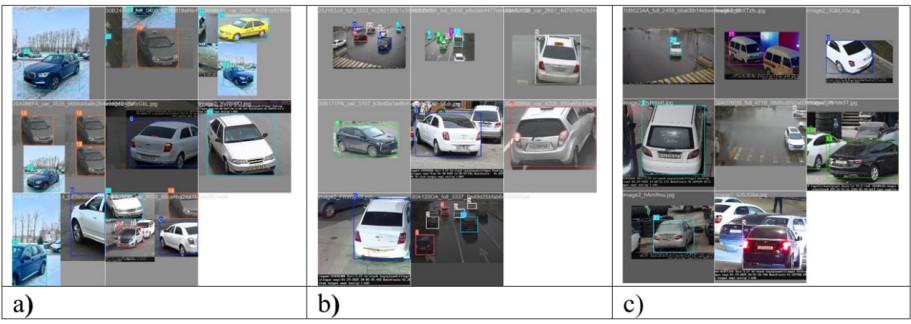

Figure 5: Qualitative results on real CCTV frames (a–c)

Figure 5 presents qualitative detection examples under real surveillance conditions. a) Static scenes and close-range views demonstrate reliable recognition across multiple vehicle models and color categories under frontal, lateral, and rear perspectives. b) Urban and parking environments illustrate stable detection performance for small-scale objects, partial occlusions, and complex backgrounds. c) Challenging scenarios—including rain, wet asphalt reflections, twilight, nighttime conditions, and strong light–shadow contrasts—highlight performance sensitivity, particularly for color estimation.

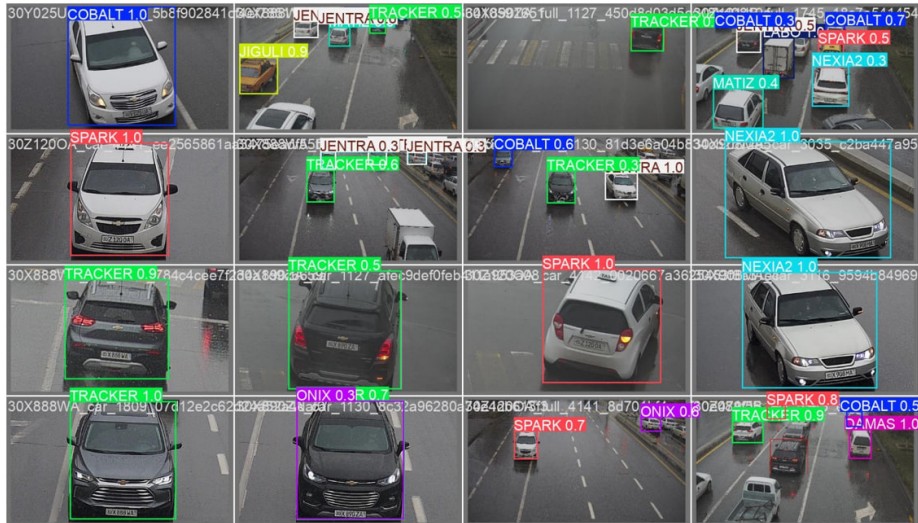

Figure 6: Highway detection results in adverse weather conditions

Fig. 6. Detection results under adverse weather conditions on highways. Despite reflections, varying scales, viewpoint changes, and low contrast, YOLOv8 maintains stable detection for most vehicle classes.

## 10.7 Performance in Real-World Conditions

Qualitative results (Figs. 5–6) demonstrate stable detection performance under challenging real-world scenarios, including rain, nighttime illumination, and long-distance detection.

The deployment pipeline supports multi-camera RTSP streams with automatic reconnection and stream management. Real-time visualization overlays are displayed through an integrated CRM platform. The system also monitors CPU/RAM/storage utilization and logs stream failures.

The implementation is fully compatible with ONNX and TensorRT export, enabling hardware-accelerated inference for real-time applications.

## 10.8 Multi-Camera CRM System

The developed system was validated in a practical setting through a Telegram bot and a web-based CRM interface. The platform enables:

Multi-camera management: RTSP stream integration, state monitoring (on/off/FPS), and automatic reconnection.

Geospatial interface: camera visualization and filtering by region, district, and GPS coordinates.

Real-time dashboard: live frame display with detected objects and confidence scores.

Integrated pipeline: unified vehicle model detection (YOLO) and color estimation (LAB+HSV).

Advanced search and filtering: by time interval, camera, model, color, and confidence threshold.

Analytical tools: class distribution analysis, PR curves, confusion matrices, and time-based statistics.

Data export and integration: CSV/XLSX/JSON export and REST/WebSocket API support.

Access control: role-based permissions and audit logging.

Performance optimization: ONNX/TensorRT acceleration and system resource monitoring (CPU, RAM, storage, network, GPU).

# 11    Conclusion

This study presents a practical and deployment-oriented pipeline for vehicle model recognition under real CCTV traffic streams in Uzbekistan. A locally curated dataset, annotated in CVAT and partitioned at the camera level (approximately 12,000 images across more than 15 classes), was prepared in COCO and YOLO formats to ensure fair evaluation.

A comparative analysis of YOLOv8-l, DETR (ResNet-101), and CenterNet demonstrated that YOLOv8-l provides the most favorable trade-off between detection accuracy, F1-score, and inference speed for real-time applications. An additional body color estimation module based on a hybrid LAB–HSV approach was integrated using detector-generated ROIs. The system was optimized for ONNX/TensorRT export and adapted for stable multi-camera RTSP deployment.

Practical validation through a Telegram bot and web-based CRM interface confirms the operational readiness of the proposed framework, enabling real-time monitoring, filtering, and analytical reporting within a unified platform. The obtained results establish a strong foundation for scalable industrial implementation in intelligent transport monitoring and urban safety systems.

# References

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
