# OpenReview forum: "Object Detection Systems for Vehicle Model and Body Color Recognition: A Comparative Study and Real-World Deployment on CCTV Data from Uzbekistan."
_mathai.club/MathAI/2026/Conference — 2026 Oral_

### Official Review · Reviewer_5Zvc · 2026-03-11
**Great study, but doesn't meet the conference theme.**

**Rating:** 6
**Confidence:** 4

**Review:**

The paper focuses on describing elaborated vehicle model and a body color recognition pipeline.

The authors consider several neural network architectures, including YOLO, DETR and CenterNet.

Great attention is paid to the organization, standardization and augmentation of the dataset, which are crucial for machine learning tasks. Acceptable classification metrics were achieved and their explicit dependence on confidence threshold values was defined.

It is worth noting that the paper does not provide a comparative analysis of the models considered. YOLOv8 was the only model whose metrics were described and interpreted. It’s also unclear why this model was chosen from the entire YOLO family.
Overall, the paper is well-organized and contains qualitative practical results.

However, It doesn't seem to suit the conference theme of mathematical foundations enough.

---

### Official Review · Reviewer_CDWr · 2026-03-11
**This paper presents a practical study on vehicle model and body color recognition tailored specifically to the automotive market of Uzbekistan (local dataset) based on YOLOv8.**

**Rating:** 6
**Confidence:** 4

**Review:**

Despite its practical value, the paper has several notable weaknesses that should be addressed.

1. There is a significant discrepancy in the reported results. The abstract states that YOLOv8-s achieves mAP@0.5 = 0.93 and an F1-score = 0.91. However, in Section 10.1 and Figure 1, the F1-confidence curve peaks at approximately 0.55, and Section 10.2 reports mAP@0.5 ≈ 0.85. This inconsistency undermines the credibility of the results. The authors must clarify which metrics are correct and explain the source of this discrepancy (e.g., different test sets, class-averaging methods, or a misprint). If the correct metrics are indeed 0.93/0.91, the paper should present them prominently and consistently.

2. The paper claims to compare YOLOv8, DETR, and CenterNet, but the analysis of DETR and CenterNet is superficial. Section 5 describes their augmentations, but Section 10 (Results) focuses almost exclusively on YOLOv8. No quantitative results (mAP, latency) are provided for DETR or CenterNet in the final evaluation. If these models were excluded due to poor performance or latency, this should be explicitly stated and justified. A true comparative study requires presenting their metrics alongside YOLOv8.

No quantitative metrics fot the hybrid LAB+HSV approach (e.g., color classification accuracy, confusion matrix for colors) are provided. The authors note that "neutral categories demonstrate stable performance," but this is vague. Given that color recognition is a core contribution, it warrants a dedicated quantitative analysis, perhaps benchmarking against a small trained classifier on a labeled color subset.

3. While the dataset size (12,000 images) and class count (15+) are mentioned, critical details are missing:
- no inter-annotator agreement metrics or quality control measures are reported.
- the number of distinct cameras and the time span over which data was collected are not specified, making it difficult to assess the dataset's diversity.

4. The paper frames its contribution around the unique Uzbekistan market, which is valid. However, it does not discuss how the approach might generalize to other regions or how the model would perform on a standard benchmark like COCO or Stanford Cars. A small experiment demonstrating cross-dataset performance (or the lack thereof) would strengthen the argument for domain-specific adaptation.

---

### Official Review · Reviewer_oxR8 · 2026-03-11
**Review of 'Object Detection Systems for Vehicle Model and Body Color Recognition: A Comparative Study and Real-World Deployment on CCTV Data from Uzbekistan'**

**Rating:** 6
**Confidence:** 4

**Review:**

The paper presents a system for vehicle model and body color recognition from CCTV traffic streams collected in Uzbekistan. A localized dataset of approximately 12,000 images across more than 15 vehicle categories is curated and annotated. The study considers several object detection architectures (YOLOv8, DETR, and CenterNet) and integrates a heuristic LAB–HSV module for vehicle color estimation. The final pipeline is designed for real-time multi-camera deployment using ONNX and TensorRT acceleration.

Strengths

1. Practical application focus: The work addresses a real-world problem in traffic monitoring and surveillance systems.
2. Localized dataset construction: The dataset reflects regional vehicle distributions, which is useful for applications where global datasets may not represent local traffic patterns.
3. Careful dataset partitioning: The camera-based train/validation/test split helps reduce potential data leakage between subsets.
4. Deployment-oriented pipeline: The paper demonstrates practical deployment considerations including RTSP streaming, ONNX export, TensorRT acceleration, and real-time monitoring tools.

Weaknesses

1. Inconsistent reporting of evaluation metrics: The abstract reports mAP@0.5 = 0.93 and F1 = 0.91, whereas the evaluation discussion presents substantially different values. In Section 10.1, the F1-confidence curve peaks around F1 = 0.55, and Section 10.2 reports mAP@0.5 = 0.85. The paper does not explain this discrepancy, making it unclear which results correspond to the final evaluation.
2. Comparative evaluation is incomplete: Although the paper states that YOLOv8, DETR, and CenterNet are evaluated (Sections 1-2 and 6), the experimental analysis in Section 10 focuses almost exclusively on YOLOv8. Quantitative results for DETR and CenterNet are not presented, which weakens the claimed comparative study.
3. Limited support for claims: The paper appears inconsistent about the YOLOv8 variant used. The abstract reports results for YOLOv8-s, while Section 6 states that YOLOv8-l was selected as the primary backbone. Clarifying which model produced the reported metrics would improve reproducibility.
4. Limited evaluation of the color recognition module: The LAB-HSV color estimation module described in Section 8 and Section 10.6 does not include quantitative evaluation metrics such as color classification accuracy or confusion statistics.
5. Dataset description lacks important details: Section 2 (Data Acquisition and Dataset Organization) provides a general overview of the dataset but omits details such as annotation verification procedures, number of cameras, and data collection period.


Thus, the paper presents a practical system for vehicle model and color recognition using localized CCTV data and demonstrates a deployment-oriented pipeline. However, several issues, particularly the inconsistency in reported evaluation metrics and the incomplete comparative analysis of the considered architectures reduce the clarity and strength of the experimental contribution. Addressing these issues and providing more detailed evaluation would significantly improve the work.

---

### Decision · Program_Chairs · 2026-03-14

**Decision:**

Accept (Oral)

**Comment:**

Dear Author(s),

On behalf of the Program Committee of the International Conference on Mathematics of Artificial Intelligence (MathAI 2026), we are pleased to inform you that your paper has been accepted for an oral presentation at MathAI 2026.

Your paper was evaluated through a rigorous two-stage review process involving both automated screening and expert review by members of the Program Committee. The reviewers recognized the quality and contribution of your work.

Presentation details:

- Format: Oral presentation (15–20 minutes + 5 minutes Q&A)
- Mode: You may present either in person (offline) at the conference venue in Sirius, Russia, or remotely via Zoom. Please indicate your preferred mode when confirming your participation.
- Conference dates: Marh 30 - April 3, 2026
- Website: https://mathai.club

Next steps:

1. Please confirm your participation and presentation mode by replying to this email mathai.club@yandex.ru no later than March 15, 2026 18:00 Moscow time.
2. If you plan to attend in person, the organizing committee will provide accommodation details separately.
3. Please prepare your final camera-ready manuscript according to the formatting guidelines available at https://mathai.club and upload it to OpenReview by March 15, 2026 18:00 Moscow time.

Should you have any questions regarding the program, logistics, or your presentation slot, please do not hesitate to contact us.

We look forward to your contribution to MathAI 2026.

With kind regards,

MathAI 2026 Program Committee
International Conference on Mathematics of Artificial Intelligence
https://mathai.club
OpenReview: https://openreview.net/group?id=mathai.club/MathAI/2026/Conference
Telegram: https://t.me/MathAI_club
Email: mathai.club@yandex.ru